# Fighting Uncertainty with Gradients: Offline Reinforcement Learning via Diffusion Score Matching

**H.J. Terry Suh**
CSAIL, MIT
Cambridge, MA 02139
hjsuh@mit.edu

**Glen Chou**[*]
CSAIL, MIT
Cambridge, MA 02139
gchou@mit.edu

**Hongkai Dai**[*]
Toyota Research Institute
Los Altos, CA 94022
hongkai.dai@tri.global

**Lujie Yang**[*]
CSAIL, MIT
Cambridge, MA 02139
lujie@mit.edu

**Abhishek Gupta**
University of Washington
Seattle, WA 98195
abhgupta@cs.washington.edu

**Russ Tedrake**
CSAIL, MIT
Cambridge, MA 02139
russt@mit.edu

**Abstract:** Gradient-based methods enable efficient search capabilities in high dimensions. However, in order to apply them effectively in offline optimization paradigms such as offline Reinforcement Learning (RL) or Imitation Learning (IL), we require a more careful consideration of how uncertainty estimation interplays with first-order methods that attempt to minimize them. We study smoothed distance to data as an uncertainty metric, and claim that it has two beneficial properties: (i) it allows gradient-based methods that attempt to minimize uncertainty to drive iterates to data as smoothing is annealed, and (ii) it facilitates analysis of model bias with Lipschitz constants. As distance to data can be expensive to compute online, we consider settings where we need amortize this computation. Instead of learning the distance however, we propose to learn its gradients directly as an oracle for first-order optimizers. We show these gradients can be efficiently learned with score-matching techniques by leveraging the equivalence between distance to data and data likelihood. Using this insight, we propose Score-Guided Planning (SGP), a planning algorithm for offline RL that utilizes score-matching to enable first-order planning in high-dimensional problems, where zeroth-order methods were unable to scale, and ensembles were unable to overcome local minima. Website: https://sites.google.com/view/score-guided-planning/home

**Keywords:** Diffusion, Score-Matching, Offline, Model-Based Reinforcement Learning, Imitation Learning, Planning under Uncertainty

## 1 Introduction

Uncertainty minimization is a central problem in offline optimization, which manifests as many different paradigms in robot learning. In offline model-based RL (MBRL [1, 2, 3, 4]), penalization of uncertainty acts as a regularizer against model bias [5, 6, 7] and prevents the optimizer from exploiting model error [8, 6, 9, 10]. In offline model-free RL, it regularizes against overestimation of the Q function [11, 12]. In addition, many imitation learning (IL) algorithms can be viewed as minimizing distribution shift from the demonstrator distribution [13] .

Despite the importance of uncertainty, statistical uncertainty quantification remains a difficult problem [14, 15, 8, 6, 16]; Gaussian processes (GPs) [6, 17, 18] rarely scale to high dimensions, and ensembles [8, 4, 19] are prone to underestimating true uncertainty [15]. As such, previous works have often taken the approach of staying near the data by maximizing data likelihood. These methods either minimize distribution shift between the optimized and data distribution for behavior regularization

---

* These authors contributed equally to this work.

7th Conference on Robot Learning (CoRL 2023), Atlanta, USA.

[20, 21, 22, 23], or the occupation measure of the optimized policy and the data distribution [11, 24, 25, 26, 27, 28, 29, 30]. Both of these require the likelihood of the data to be estimated.

A promising direction for estimating the data likelihood is to leverage techniques from likelihood-based generative modeling, such as variational autoencoders (VAE) [31, 32, 33], generative adversarial networks (GAN) [34, 35, 36, 13], and flow-based models [25, 37]. Yet, these prior works have shown that training density models to generate accurate likelihoods can be challenging, especially for high-dimensional data. While gradient-based methods have good scalability properties that make them desirable for tackling offline optimization problems in this high-dimensional regime, the effect of incorrect likelihoods are further exacerbated in this setting as they lead gradient-based methods into spurious local minima. Thus, we ask in this work: can we design gradient-based offline optimization methods that encourage data likelihood, without explicit generative modeling of likelihoods?

Our key insight is that in order to maximize the data likelihood with gradient-based methods, we do not need access to the likelihood itself. Rather, having access to a first-order oracle (gradients), known as the *score function*, is sufficient. We claim that directly utilizing the score function has two benefits compared to likelihood-based modeling. i) First, recent breakthroughs in score-based modeling [38, 39, 40] show that the score function is considerably easier to estimate with score-matching techniques [40, 39], as it bypasses estimation of the partition function that is required for computation of exact likelihoods [40, 41]. ii) In addition, we show that score matching with annealed perturbations [40] gives gradients that stably drive decision variables to land exactly on data when uncertainty is minimized with gradient-based optimization, a property we term *data stability*. We demonstrate this by showing that the negative log likelihood of the perturbed empirical distribution, whose gradients score-matching estimates, is equivalent to a softened distance to data [42].

Furthermore, we ask: when, and why, would approaches that penalize distance to data surpass the ensemble method of statistical uncertainty quantification [4, 9]? We show that unlike empirical variance among ensembles, we can relate how much smoothed distance to data underestimates true uncertainty with the Lipschitz constant, for which we can use statistical estimation to put confidence bounds [18], or utilize structured domain knowledge [43]. Moreover, we show that ensembles do not necessarily have the data stability property due to statistical noise; therefore, optimizing for ensemble variance can easily lead to local minima away from data in gradient-based optimization.

To put our theory into a practical algorithm, we propose Score-Guided Planning (SGP), a gradient-based algorithm that estimates gradients of the log likelihood with score matching, and solves uncertainty-penalized offline optimization problems that additively combine the cumulative reward and the log likelihood of data *without* any explicit modeling of likelihood. SGP enables stable uncertainty minimization in high-dimensional problems, enabling offline MBRL to scale even to pixel action-spaces. We validate our theory on empirical examples such as the cart-pole system, the D4RL benchmark [44], a pixel-space single integrator, and a box-pushing task [45] on hardware.

## 2 Preliminaries

**Offline Model-Based Optimization.** We first introduce a setting of *offline model-based optimization* [46]. In this setting, we aim to find $x$ that minimizes an objective function $f : \mathbb{R}^n \to \mathbb{R}$, but are not directly given access to $f$; instead, we have access to $x_i \sim p(x)$, and their corresponding values $f(x_i)$, such that the dataset consists of $\mathcal{D} = \{(x_i, f(x_i))\}$. Denoting $\hat{p}(x; \mathcal{D})$ as the empirical distribution corresponding to dataset $\mathcal{D}$, offline model-based optimization solves

$$\min_x f_{\theta^*}(x) \quad \text{s.t.} \quad \theta^* = \arg\min_\theta \mathbb{E}_{x \sim \hat{p}(x; \mathcal{D})} \left[ \|f_\theta(x) - f(x)\|^2 \right]. \tag{1}$$

In words, we minimize a surrogate loss $f_\theta(x)$, where we choose $\theta$ as the solution to empirical risk minimization of matching $f$ given the data. Denoting $x^* = \arg\min_x f_{\theta^*}(x)$, one measure of performance of this procedure is error at optimality, $\|f(x^*) - f_\theta(x^*)\|$.

**Uncertainty Penalization.** The gap $\|f(x^*) - f_\theta(x^*)\|$ potentially can be large if $f_\theta$ fails to approximate $f$ correctly at $x^*$, which is likely if $x^*$ is out-of-distribution (o.o.d.). To remedy this,

previous works [8, 46, 47] have proposed adding a loss term that penalizes o.o.d. regions. We denote this penalized objective as

$$\bar{f}_\theta(x) := f_\theta(x) + \beta\mu^2(x), \tag{2}$$

where $\beta \in \mathbb{R}_{\geq 0}$ is some weighting parameter, and $\mu(x)$ is some notion of uncertainty. Intuitively this restricts the choice of optimal $x$ to the training distribution used to find $\theta^*$, since these are the values that can be trusted. If the uncertainty metric overestimates the true uncertainty, $\|f(x) - f_\theta(x)\| \leq \mu(x)$, it is possible to bound the error at optimality directly using $\mu(x^*)$.

**Offline Model-Based RL.** While the offline model-based optimization problem described is a one-step problem, the problem of offline model-based RL (MBRL) involves sequential decision making using an offline dataset. In offline MBRL, we are given a dataset $\mathcal{D} = \{(x_t, u_t, x_{t+1})_i\}$ where $x \in \mathbb{R}^n$ denotes the state, $u \in \mathbb{R}^m$ denotes action, $t$ is the time index and $i$ is the sample index. Again denoting $\hat{p}(x_t, u_t, x_{t+1}; \mathcal{D})$ as the empirical distribution corresponding to $\mathcal{D}$, and introducing $\mu(x_t, u_t)$ as a state-action uncertainty metric [8], uncertainty-penalized offline MBRL solves

$$
\begin{aligned}
\max_{x_{1:T}, u_{1:T}} \quad & \textstyle\sum_{t=1}^T r_t(x_t, u_t) - \beta\mu^2(x_t, u_t) \\
\text{s.t.} \quad & x_{t+1} = f_{\theta^*}(x_t, u_t) \ \forall t, \\
& \theta^* = \arg\min_\theta \mathbb{E}_{(x_t, u_t, x_{t+1}) \sim \hat{p}} \left[ \|x_{t+1} - f_\theta(x_t, u_t)\|^2 \right].
\end{aligned}
\tag{3}
$$

In words, we first approximate the transition dynamics from data, then solve the uncertainty-penalized optimal control problem assuming the learned dynamics. Previous works have used ensembles [8, 9, 48], or likelihood-based generative models of data [25] to estimate this uncertainty.

The resulting open-loop planning problem can either be solved with first-order methods [49, 50], or zeroth-order sampling-based methods [45, 9] such as CEM [51] or MPPI [52]. While insights from stochastic optimization [53] tell us that first-order methods are more favorable in high dimensions and longer horizons as the variance of zeroth-order methods scale with dimension of decision variables [54], first-order methods require careful consideration of how amenable the uncertainty metric is to gradient-based optimization.

## 3 Distance to Data as a Metric of Uncertainty

In order to find a metric of uncertainty that is amenable for gradient-based offline optimization, we investigate smoothed distance to data as a candidate. We show that this metric allows us to drive iterates of gradient-based methods that minimize uncertainty to land on data as smoothing is annealed. In addition, it allows us to quantify how much we underestimate true uncertainty by using the Lipschitz constant of error. All proofs for theorems in this section are included in Appendix A.

### 3.1 Properties of Distance to Data

We first formally define our proposed metric of uncertainty for offline model-based optimization.

**Definition 3.1** (Distance to Data). Consider a dataset $\mathcal{D} = \{x_i\}$ and an arbitrary point $x \in \mathbb{R}^n$. The standard squared distance from $x$ to the set $\mathcal{D}$ can be written as $d(x; \mathcal{D})^2 = \min_{x_i \in \mathcal{D}} \frac{1}{2}\|x - x_i\|^2$. We define smoothed distance to data using a smoothed version of this standard squared distance,

$$d_\sigma(x; \mathcal{D})^2 := \mathsf{Softmin}_\sigma \tfrac{1}{2}\|x - x_i\|^2 + C = -\sigma^2 \log\left[\textstyle\sum_i \exp\left[-\tfrac{1}{2\sigma^2}\|x - x_i\|^2\right]\right] + C, \tag{4}$$

where $C$ is some constant to ensure positiveness of $d_\sigma(x; \mathcal{D})^2$, and $\sigma > 0$, also known as the *temperature parameter*, controls the degree of smoothing with $\sigma \to 0$ converging to the true min.

The motivation for introducing smoothing is to make the original non-smooth distance metric more amenable for gradient-based optimization [55, 56]. We now consider benefits of using $\mu(x) = d_\sigma(x; \mathcal{D})$ as our uncertainty metric, and show that as the smoothing level is annealed down, minimizing this distance allows us to converge to the points in the dataset.

**Proposition 1** (Data Stability). Consider a monotonically decreasing sequence $\sigma_k$ such that $\sigma_k \to 0$, and denote $x_k^n$ as the $n^{th}$ gradient descent iteration of $\min_x d_{\sigma_k}(x; \mathcal{D})^2$. Then, almost surely with

random initialization and appropriate step size, we have

$$\lim_{k\to\infty}\lim_{n\to\infty} x_k^n \in \mathcal{D}. \tag{5}$$

We note that empirical variance among ensembles [8] is prone to having local minima away from data due to statistical variations, especially with small number of ensembles, which makes them unreliable for gradient-based optimization. Next, we show that it is possible to analyze how much softened distance to data underestimates true uncertainty using the Lipschitz constant of the model bias $L_e$.

**Proposition 2** (Lipschitz Bounds). Let $L_e$ be the local Lipschitz [57] constant of the true error (bias) $e(x) \coloneqq \|f(x) - f_\theta(x)\|_2$ valid over $\mathcal{Z} \subseteq \mathcal{X}$, where $\mathcal{X}$ is the domain of the input $x$. Then, $e(x)$ is bounded by

$$e(x) \le e(x_c) + \sqrt{2}L_e\sqrt{d_\sigma(x;\mathcal{D})^2 + C_2}, \tag{6}$$

for all $x \in \mathcal{Z}$, where $x_c \coloneqq \arg\min_{x_i\in\mathcal{D}} \frac{1}{2}\|x - x_i\|_2^2$, i.e., the closest data-point, and $C_2 = \sigma^2 \log N - C$, where $C$ is defined in (4).

In general, it is difficult to obtain $L_e$ in the absence of more structured knowledge of $f$. However, it is possible to obtain confidence bounds on $L_e$ using statistical estimation with pairwise finite slopes $\|e(x_i) - e(x_j)\|/\|x_i - x_j\|$ within the dataset [58, Ch. 3] [18] . We believe this offers benefits over ensembles as characterizing the convergence of neural network weights with randomly initialized points is far more complex to analyze. We compare distance to data to other uncertainty metrics in Figure 1 in simple 1D offline model-based optimization, where we show that ensembles [8, 9] have local minima outside of data, and unpredictably underestimates uncertainty due to model bias.

## 3.2 Estimating Gradients of Distance to Data with Score Matching

Although we have shown benefits of smoothed distance to data as an uncertainty metric amenable for gradient-based optimization, it is costly to compute at inference time as we need to iterate through the entire dataset. At the cost of losing guarantees of exact computation in Section 3.1, we consider ways to amortize this computation with function approximation. We first show the equivalence of the smoothed distance-to-data to the negative log likelihood of the *perturbed empirical distribution* [40], which applies randomized smoothing [55, 56] to the empirical distribution $\hat{p}(x;\mathcal{D})$.

**Definition 3.2** (Perturbed Empirical Distribution). Consider a dataset $\mathcal{D} = \{x_i\}$ and its corresponding empirical distribution $\hat{p}(x;\mathcal{D})$. We define $p_\sigma(x;\mathcal{D})$ as the noise-perturbed empirical distribution,

$$p_\sigma(x';\mathcal{D}) \coloneqq \int \hat{p}(x;\mathcal{D})\mathcal{N}(x';x,\sigma^2\mathbf{I})dx = \frac{1}{N}\sum_{x_i\in\mathcal{D}}\mathcal{N}(x';x_i,\sigma^2\mathbf{I}). \tag{7}$$

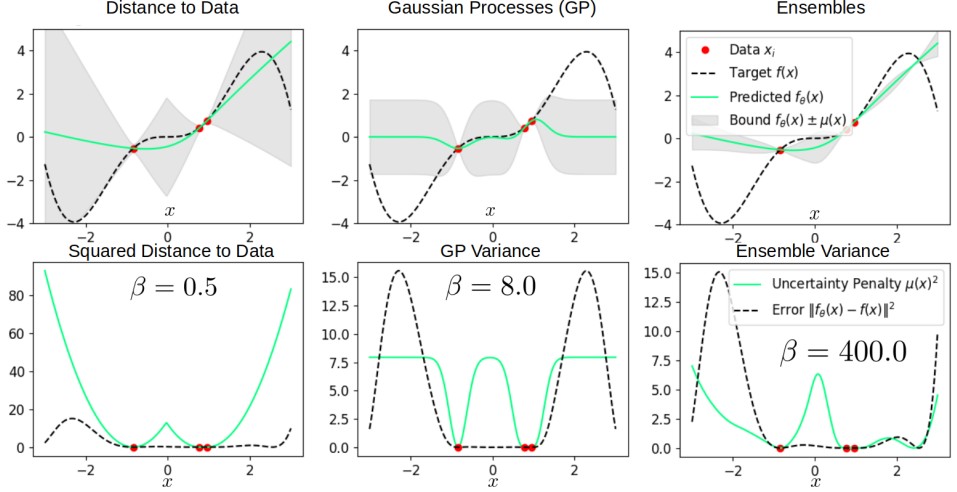

Figure 1: Comparison of different uncertainty metrics. Top row: Visualization of distance to data against GPs and ensembles with $M = 2$. Bottom row: Visualization of the penalty $\mu(x)^2$. All the metrics underestimate uncertainty to varying degrees, but distance to data can be more amenable for analysis; as we increase samples, distance to data will be able to bound the true uncertainty more closely by estimating Lipschitz constants using pairwise slopes in the dataset. In addition, distance to data shows more stability to data while ensembles have local minima outside of data.

**Proposition 3.** The negative log-likelihood of the perturbed empirical distribution $p_\sigma(x)$ is equivalent to smoothed distance to data by a factor of $\sigma^2$, up to some constant that does not depend on $x$,

$$-\sigma^2 \log p_\sigma(x; \mathcal{D}) = d_\sigma(x; \mathcal{D})^2 + C(N, n, \sigma). \tag{8}$$

This connection to the perturbed empirical distribution allows us to use generative modeling tools that randomly perturb data [59, 40, 39], such as denoising autoencoders. However, computing likelihoods directly have shown to be difficult for high-dimensional data [40]. Additionally, even if such a likelihood-based generative model can estimate likelihoods with low error, there are no guarantees that its gradients will also be of low error (Appendix A.4), which defeats our purpose of finding a metric amenable to gradient-based optimization. Thus, we propose to estimate the gradients of the perturbed empirical distribution (score function) directly [40, 39], which have shown promising performance in generative modeling [38] as it bypasses the estimation of the partition function.

Estimating the score function is a process known as *score-matching*. Following [40], we introduce *noise-conditioned score function* $s(x, \sigma) := \nabla_x \log p_\sigma(x; \mathcal{D})$ [40] and aim to optimize the following objective given some decreasing sequence of annealed smoothing parameters $\sigma_k$,

$$\min_\theta \sum_k \sigma_k^2 \mathbb{E}_{x \sim p_{\sigma_k}(x; \mathcal{D})} \left[ \| s_\theta(x, \sigma_k) - \nabla_x \log p_{\sigma_k}(x; \mathcal{D}) \|^2 \right], \tag{9}$$

which has been shown to be equivalent to the denoising-score-matching loss [59]. Compared to explicitly computing $\nabla_x \log p_{\sigma_k}(x; \mathcal{D})$ which would require iterating through the entire dataset, the denoising loss allows us to learn the score function using batches of data.

$$\min_\theta \sum_k \sigma_k^2 \mathbb{E}_{\substack{x \sim \hat{p}(x; \mathcal{D}) \\ x' \sim \mathcal{N}(x'; x, \sigma_k^2 \mathbf{I})}} \left[ \| s_\theta(x'; \sigma_k) + \sigma_k^{-2}(x' - x) \|^2 \right]. \tag{10}$$

# 4   Planning with Gradients of Data Likelihood

Having illustrated benefits of using smoothed distance to data as an uncertainty metric, we now turn to the sequential decision-making setting of offline MBRL, where we use notation from Section 2.

**Offline MBRL with Data Likelihood.** Consider a learned dynamics model $f_\theta$ from the dataset $\mathcal{D} = \{(x_t, u_t, x_{t+1})_i\}$, as well as the perturbed empirical distribution $p_\sigma(x_t, u_t; \mathcal{D})$ of the $(x_t, u_t)$ pairs in the dataset $\mathcal{D}$. Then, given some sequence of rewards $r_t$, we consider the following planning problem of maximizing both the reward and the likelihood of data,

$$\max_{x_{1:T}, u_{1:T}} \quad \sum_{t=1}^T r_t(x_t, u_t) + \beta \sigma^2 \sum_{t=1}^T \log p_\sigma(x_t, u_t; \mathcal{D}) \tag{11}$$
$$\text{s.t.} \quad x_{t+1} = f_\theta(x_t, u_t) \quad \forall t \in [1, T].$$

Note that $-\sigma^2 \log p_\sigma(x_t, u_t)$ is equivalent to smoothed distance to data $d_\sigma(x_t, u_t; \mathcal{D})^2$ as an objective, and acts as an uncertainty penalty, preventing the optimizer from exploiting o.o.d. solutions.

**Score-Guided Planning (SGP).** Given a differentiable $r_t$, we propose a single-shooting algorithm that rolls out $f_\theta$ and computes gradients of Equation (11) with respect to the input trajectory. We first denote the sensitivity of the state and input at time $t$ $(x_t, u_t)$ with respect to the input at time $j$ as $\partial x_t / \partial u_j$ and $\partial u_t / \partial u_j$. Then, the gradients of Equation (11) can be expressed using this sensitivity,

$$\frac{\partial}{\partial u_j} \left[ \sum_{t=1}^T r_t(x_t, u_t) + \beta \sigma^2 \sum_{t=1}^T \log p_\sigma(x_t, u_t; \mathcal{D}) \right]$$
$$= \sum_{t=1}^T \left[ \frac{\partial r_t}{\partial x_t} \frac{\partial x_t}{\partial u_j} + \frac{\partial r_t}{\partial u_t} \frac{\partial u_t}{\partial u_j} + \beta \sigma^2 \left[ \frac{\partial \log p_\sigma(x_t, u_t)}{\partial x_t} \frac{\partial x_t}{\partial u_j} + \frac{\partial \log p_\sigma(x_t, u_t)}{\partial u_t} \frac{\partial u_t}{\partial u_j} \right] \right]. \tag{12}$$

Note that $\partial \log p_\sigma(x_t, u_t) / \partial x_t$ and $\partial \log p_\sigma(x_t, u_t) / \partial u_t$ can be obtained using the noise-conditioned score estimator $s_\theta(x, u, \sigma)$ in Section 3.2, where we extend the domain to include both $x$ and $u$. After using reverse-mode autodiff (Appendix B.1) to compute the gradient, we call optimizers that accept gradient oracles, such as Adam [50]. We also note that this gradient computation can be extended

to feedback policy gradients in Appendix B.5. Finally, we train a noise-conditioned score function for some sequence $\sigma_k$, and anneal the noise-level during optimization following [40]. SGP is also described with an algorithm block in Appendix B.3.

**Related Methods**. LDM, COMBO, and DOGE [25, 12, 60] are closely-related offline methods that penalize for data likelihood or distance to data. Many IL methods [41] are also related, as SGP with zero rewards can be used to imitate demonstration data by maximizing data likelihood [13] (Appendix B.6). In particular, AIRL [35] maximizes a rewardless version of our objective using GANs[34]. However, these methods primarily rely on likelihood-based generative models, and do not consider the interplay of generative modeling with gradient-based optimization. Diffuser [26, 61] shares similar methodologies with SGP, and solves a variant of Equation (11) with a quadratic-penalty-based approximation of direct transcription (Appendix B.4), which comes with benefits of numerical stability for long horizons [62, exercise 10.1], and robustness to sparse rewards [63].

# 5    Empirical Results

We now test our proposed algorithm (SGP) empirically and show that it is an effective method for offline optimization that has scalable properties in high-dimensional problems by leveraging gradient-based optimization. All details for the included environments are presented in Appendix C.

**Cart-pole System with Learned Dynamics.**    We apply our method to swing up a cart-pole model, which undergoes a long duration ($T = 60$) dynamic motion, which directly translates to high number of decision variables in single shooting. We first show the effects of model bias in in Fig. 2.C, where vanilla MBRL plans a trajectory that leaves the region (red box) of training data. In contrast, SGP keeps the trajectory within the training distribution (Fig.2.A). We further compare our method against the baselines of ensembles (Fig.2.B,E) and CEM [51] (Fig.2.D,E). To fairly compare CEM against SGP, we additionally train a data distance estimator that predicts the likelihoods directly based on explicit computation of the softmin distance in Equation (4) (Appendix C).

We observe that the convergence of CEM is much slower than first-order methods (Fig.2.F). However, we surprisingly obtain more asymptotic performance with CEM rather than using Adam for ensembles - we believe this signifies presence of local minima in gradient-based minimization of ensemble variance, as opposed to CEM which has some stochastic smoothing that allows it to escape local minima [54]. Finally, we note that unlike score functions, distance to data is considerably more difficult to train as we need to loop through the entire dataset to compute one sample. As a result, it is costly to train and we were unable to train it to good performance in a reasonable amount of time.

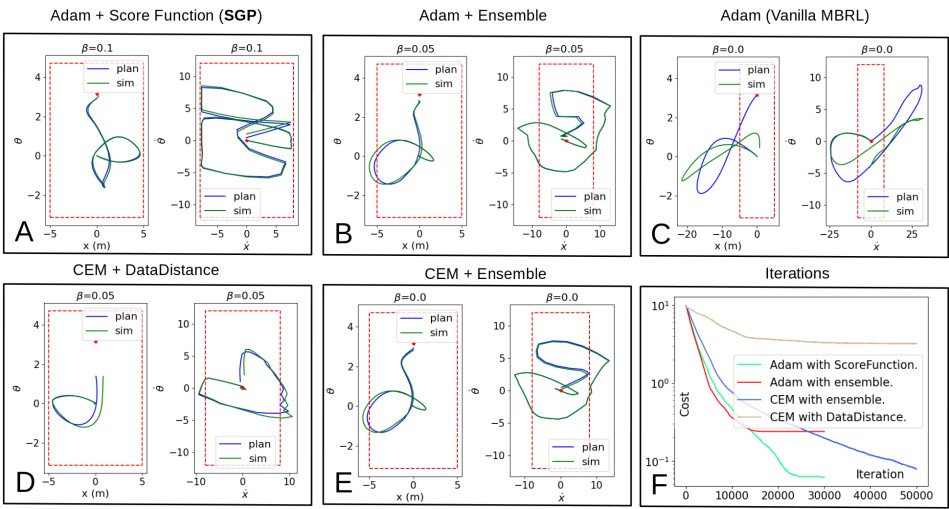

Figure 2: Experiments for the cart-pole system, where blue is the hallucinated plan and green is the actual rollout, and the red box is the regime of collected data. Note that SGP (**A**) results in a successful swingup while MBRL (**C**) fails. Compared to other baselines such as **B,D,E**, our method achieves much lower true cost with faster convergence (**F**).

| Dataset | Env | BC | MOPO | MBOP | AWAC | CQL | Diffuser | Ours |
|---|---|---|---|---|---|---|---|---|
| Random | halfcheetah | 2.1 | **35.4**±2.5 | 6.3±4.0 | 2.2 | **35.4** | - | 31.3±0.5 |
| Random | hopper | 1.6 | **11.7**±0.4 | 10.8±0.3 | 9.6 | 10.8 | - | **12.3**±1.3 |
| Random | walker2d | 9.8 | 13.6±2.6 | 8.1±5.5 | 5.1 | 7.0 | - | **15.6**±7.1 |
| Medium | halfcheetah | 42.6 | 42.3±1.6 | 44.6±0.8 | 37.4 | 44.0 | 42.8±0.3 | **46.6**±2.8 |
| Medium | hopper | 52.9 | 28.0±12.4 | 48.8±26.8 | **72.0** | 58.5 | **74.3**±1.4 | 60.6±5.8 |
| Medium | walker2d | 75.3 | 17.8±19.3 | 41.0±29.4 | 30.1 | 72.5 | **79.6**±0.6 | 36.2±6.8 |
| Med-Exp | halfcheetah | 55.2 | 63.3±38.0 | **105.9**±17.8 | 36.8 | 91.6 | 88.9±0.3 | 61.6±2.4 |
| Med-Exp | hopper | 52.5 | 23.7±6.0 | 55.1±44.3 | 80.9 | **105.4** | 103.3±1.3 | **109.2**±0.3 |
| Med-Exp | walker2d | **107.5** | 44.6±12.9 | 70.2±36.2 | 42.7 | 108.8 | **106.9**±0.2 | 103.0±2.8 |

Table 1: Comparing performance of our algorithm on the D4RL [44] dataset. Similar to [26, 68], we bold-face 95% of max performance. Ours is averaged from 5 trials. We emphasize that we outperform ensembles (MOPO) in many of the tasks.

**D4RL Benchmark.** To evaluate our method against other methods on a standard benchmark, we use the MuJoCo [64] tasks in the D4RL [44] dataset with three different environments and sources of data. To turn our planner into a controller, we solve the planning problem with some finite horizon, rollout the first optimal action $u_1^*$, and recompute the plan in a standard model-predictive control (MPC) [65] fashion. We compare against methods such as Behavior Cloning (BC) [22, 21], MOPO [8], MBOP [66], CQL [23], AWAC [67], and Diffuser [26]. Our results demonstrate that our algorithm performs comparably to other state-of-the-art methods. On many of the tasks, we demonstrate better performance compared to MOPO [8] which uses ensembles for uncertainty estimation, while requiring less memory. This empirically supports our proposed benefits of using score matching for gradient-based offline MBRL. In addition, we outperform BC in many tasks, illustrating that we achieve better performance than pure imitation learning by incorporating rewards.

**Pixel-Based Single Integrator.** Pixel spaces have long been a challenge for MBRL due to the challenges of reliably controlling from images [69], learning pixel-space dynamics, and combating the resulting model-bias [2, 30, 70, 71]. To show scalability of our method, we present a $32 \times 32$ pixel-based single integrator environment where the action space is defined in pixel space, similar to a spatial action map [72]; see Fig. 3 for visuals. Our planning problem requires proposing a sequence of $32 \times 32$ control images, which minimizes cost (goal-reaching while minimizing running cost) when rolled out through the dynamics (see Appendix C for details and numerical results). We demonstrate that SGP capably mitigates model bias, generating plans consisting of plausible observation/controls near the data; thus, these plans are closely followed when executed open-loop under the true dynamics (Fig.3.A). When ignoring the effect of model bias and planning naïvely with MBRL (Fig.3.B), i.e., setting $\beta = 0$, the resulting trajectory exploits the subtleties of the chosen cost function, planning an unrealistic trajectory which leads to high cost at runtime. Also, we show that ensembles are unable to stably converge back to the data manifold of images seen during training, instead getting trapped in local minima, planning unrealistic trajectories which translate to poor runtime rollouts on the true system (Fig.3.C). Moreover, due to the high dimensionality of this action space ($u_t \in \mathbb{R}^{32^2}$), we see that zeroth-order methods, such as CEM [51], lead to very low convergence rates (Fig.3.D) due to the large number of decision variables [53, 54]; again, this leads to high cost seen at runtime.

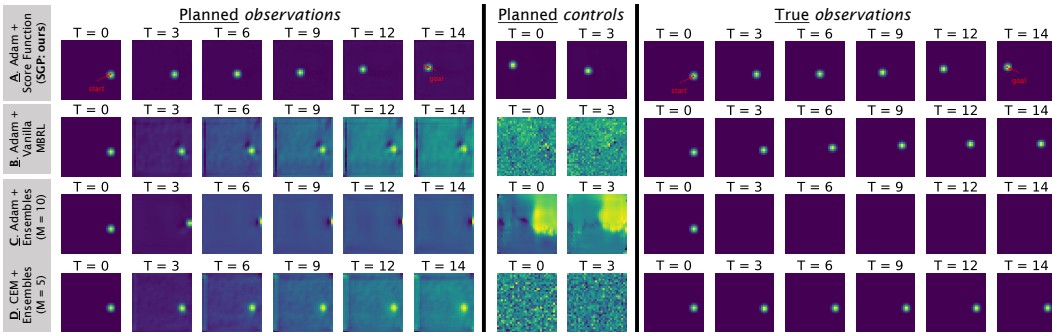

Figure 3: Planned (left) and true (right) rollouts of observations for the pixel single-integrator with pixel action space (center). Note that SGP (**A**) significantly outperforms Vanilla MBRL (**B**), Adam + Ensembles (**C**), and CEM + Ensembles (**D**), which are unable to combat model bias during rollout.

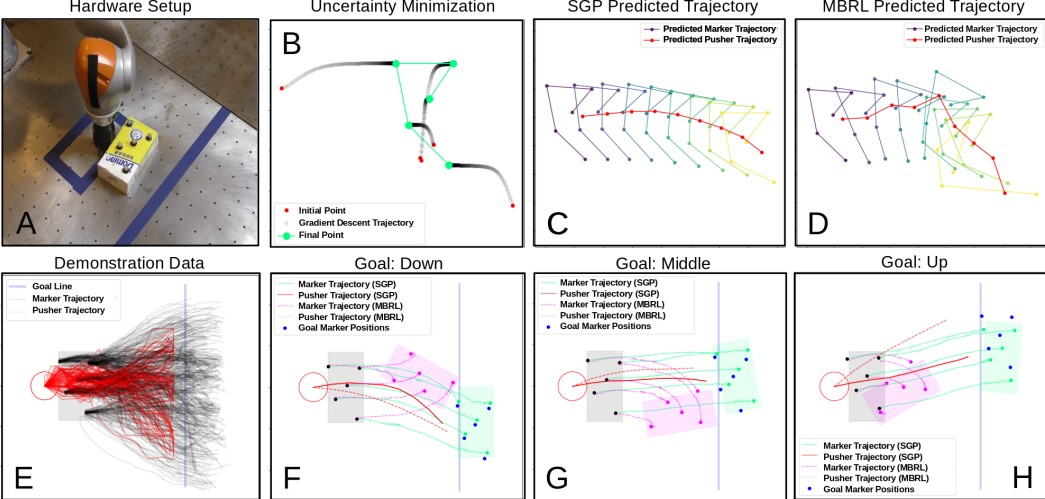

Figure 4: Visualization of the box-pushing experiment in **A**, where data was collected from 100 demonstrations illustrated in **E**. We show that SGP successfully stabilizes markers back to their implicit constraints (**B**), which allows better predictions (**C**) than vanilla MBRL (**D**). We additionally use the reward to change the goal into three different regions of down (**F**), middle (**G**), and up (**H**), where vanilla MBRL fails, and behavior cloning is not capable of.

**Box-Pushing with Learned Marker Dynamics.** To validate our method on the data-scarce regime of hardware, we prepare a box-pushing task from [45] where we leverage the quasistatic assumption [73] and treat the positions of the motion capture markers as if they are states. An interesting feature of this setup is that the markers implicitly live on a constraint manifold where the distance between each marker is fixed; we ask if using score matching can stabilize the rollouts of prediction to obey this implicit constraint, similar to how diffusion stabilizes back to the data manifold [42]. To test this, we collect about 100 demonstrations of the box being pushed to different positions, which lead to 750 samples of data tuples $(x_t, u_t, x_{t+1})$. We aim to show that i) SGP can enforce implicit constraints within the data, ii) distance to data acts as a successful uncertainty metric in data-scarce offline MBRL, and iii) we can use the reward to change the task from imitation data.

Our results in Fig.4 demonstrate that SGP successfully imposes implicit constraints on the data. Not only does minimization of uncertainty in the absence of rewards result in stabilization to the marker position constraints (Fig.4.B), but the rollouts also become considerably more stable when we use the distance to data penalty (Fig.4.C). In contrast, MBRL with $\beta = 0$ destroys the keypoint structure as dynamics are rolled out, resulting in suboptimal performance (Fig.4.D). Finally, we demonstrate through Fig.4.F,G,H that we can use rewards to show goal-driven behavior to various goals from a single set of demonstration data, which behavior cloning is not capable of.

## 6   Conclusion and Discussion of Limitations

We proposed SGP, which is a *first-order* offline MBRL planning algorithm that learns gradients of distance to data with score-matching techniques, and solves planning problems that jointly maximize reward and data likelihood. Through empirical experiments, we showed that SGP beats baselines of zeroth-order methods and ensembles, has comparable performance with state-of-art offline RL algorithms, and scales to pixel-space action spaces with up to $15,360$ decision variables for planning.

We conclude with listing some limitations of our approach. Unlike ensembles, our method by construction discourages extrapolation, which can be a limitation when the networks jointly recover meaningful inductive bias. We also believe that computation is a current bottleneck for realtime application of our MPC, which can take between 1-2 seconds per iteration due to gradient computations. Finally, we have not investigated the performance of SGP under aleatoric uncertainty [48].

**Acknowledgments**

This work was funded by Amazon.com Services LLC; Award No. PO #2D-06310236, and Office of Naval Research (ONR); Award No. N00014-22-1-2121. We would like to thank Pulkit Agrawal and Max Simchowitz for helpful discussions on the paper.

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

# A  Proofs

## A.1  Proof for Proposition 1

For each $k$, we denote the stationary point of gradient descent as $x_k^* = \lim_{n\to\infty} x_k^n$. Using the first-order optimality condition at the stationary point, we know that

$$
\begin{aligned}
0 &= \nabla_x d_{\sigma_k}(x; \mathcal{D})^2 \big|_{x=x_k^*} \\
&= -\nabla_x \sigma_k^2 \log\left[\sum_i \exp\left[-\tfrac{1}{2\sigma_k^2}\|x - x_i\|^2\right]\right]\Bigg|_{x=x_k^*} \\
&= \frac{\sigma_k^2 \sum_i \exp\left[-\tfrac{1}{2\sigma_k^2}\|x - x_i\|^2\right]\sigma_k^{-2}(x - x_i)}{\sum_i \exp\left[-\tfrac{1}{2\sigma_k^2}\|x - x_i\|^2\right]}\Bigg|_{x=x_k^*}.
\end{aligned}
\tag{13}
$$

We then have that

$$
\begin{aligned}
\sum_i \exp\left[-\tfrac{1}{2\sigma_k^2}\|x_k^* - x_i\|^2\right](x_k^* - x_i) &= 0 \\
\big(\sum_i \exp\left[-\tfrac{1}{2\sigma_k^2}\|x_k^* - x_i\|^2\right]\big)x_k^* &= \sum_i \exp\left[-\tfrac{1}{2\sigma_k^2}\|x_k^* - x_i\|^2\right] x_i \\
x_k^* &= \sum_i \frac{\exp\left[-\tfrac{1}{2\sigma_k^2}\|x_k^* - x_i\|^2\right]}{\sum_n \exp\left[-\tfrac{1}{2\sigma_k^2}\|x_k^* - x_n\|^2\right]} x_i.
\end{aligned}
\tag{14}
$$

Let $x_j \in \operatorname{argmin}_{x_i \in \mathcal{D}} \|x_k^* - x_i\|^2$ and $\Delta_i(\sigma_k) = \sigma_k^{-2}(\|x_k^* - x_i\|^2 - \|x_k^* - x_j\|^2)$.

**Lemma 4.** *When there is a unique minimizer $x_j$, $\lim_{k\to\infty} x_k^* = x_j$.*

*Proof.* Since $x_j$ is the unique closest data point to $x_k^*$, $\Delta_i > 0$ for $i \neq j$ and $\Delta_j = 0$. With the monotonically decreasing $\sigma_k \to 0$, we know that

$$
\lim_{k\to\infty} \Delta_i(\sigma_k) = \begin{cases} 0 & \text{if } i = j \\ \infty & \text{otherwise.} \end{cases}
\tag{15}
$$

Therefore,

$$
\lim_{k\to\infty} \exp\left[-\tfrac{1}{2}\Delta_i(\sigma_k)\right] = \begin{cases} 1 & \text{if } i = j \\ 0 & \text{otherwise.} \end{cases}
\tag{16}
$$

We then obtain

$$
\lim_{k\to\infty} \frac{\exp\left[-\tfrac{1}{2\sigma_k^2}(\|x_k^* - x_i\|^2 - \|x_k^* - x_j\|^2)\right]}{\sum_n \exp\left[-\tfrac{1}{2\sigma_k^2}(\|x_k^* - x_n\|^2 - \|x_k^* - x_j\|^2)\right]} = \lim_{k\to\infty} \frac{\exp\left[-\tfrac{1}{2}\Delta_i(\sigma_k)\right]}{\sum_{n\neq j} \exp\left[-\tfrac{1}{2}\Delta_n(\sigma_k)\right] + 1}
\tag{17}
$$
$$
= \begin{cases} 1 & \text{if } i = j \\ 0 & \text{otherwise.} \end{cases}
$$

Therefore, by combining (14) and (17), we have that

$$
\lim_{k\to\infty} x_k^* = x_j.
\tag{18}
$$

$\square$

**Lemma 5.** *When there are multiple minimizers $x_{j_1}, \ldots, x_{j_m} \in \operatorname{argmin}_{x_i \in \mathcal{D}} \|x_k^* - x_i\|^2$, $\lim_{k\to\infty} x_k^* = \frac{1}{m}\sum_{l=1}^m x_{j_l}$.*

*Proof.* In contrast to (17) for the unique $x_j$, we have that

$$
\begin{aligned}
\lim_{k\to\infty} &\frac{\exp\left[-\frac{1}{2}(\|x_k^* - x_i\|^2 - \|x_k^* - x_{j_1}\|^2)\right]}{\sum_n \exp\left[-\frac{1}{2}(\|x_k^* - x_n\|^2 - \|x_k^* - x_{j_1}\|^2)\right]} \\
&= \lim_{k\to\infty} \frac{\exp\left[-\frac{1}{2}\Delta_i(\sigma_k)\right]}{\sum_{n\notin\{j_1,\ldots,j_m\}} \exp\left[-\frac{1}{2}\Delta_n(\sigma_k)\right] + m} \\
&= \begin{cases} \frac{1}{m} & \text{if } i \in \{j_1,\ldots,j_m\} \\ 0 & \text{otherwise.} \end{cases}
\end{aligned}
\tag{19}
$$

Therefore, $x_k^*$ converges to the geometric center of all the minimizers:

$$
\lim_{k\to\infty} x_k^* = \frac{1}{m}\sum_{l=1}^m x_{j_l}.
\tag{20}
$$

$\square$

**Lemma 6.** *For a local minimizer $x_k^*$, $x_j$ is unique; for a local maximizer or saddle point $x_k^*$, $x_j$ is not unique.*

*Proof.* We first prove that $x_j$ is unique if $x_k^*$ is a local minimizer by contradiction. If there are multiple minimizers $x_{j_1},\ldots,x_{j_m} \in \operatorname{argmin}_{x_i \in \mathcal{D}} \|x_k^* - x_i\|^2$, we have that

$$
\begin{aligned}
\lim_{k\to\infty} d_{\sigma_k}(x_k^*;\mathcal{D})^2 &= \lim_{k\to\infty} -\sigma_k^2 \log\left[\sum_i \exp\left[-\frac{1}{2\sigma_k^2}\|x_k^* - x_i\|^2\right]\right] \\
&= -\log\left[\lim_{k\to\infty}\left(\sum_i \exp\left[-\frac{1}{2\sigma_k^2}\|x_k^* - x_i\|^2\right]\right)^{\sigma_k^2}\right] \\
&= \min_i \lim_{k\to\infty} \|x_k^* - x_i\|^2 \\
&= \lim_{k\to\infty} \|x_k^* - x_{j_1}\|^2 \\
&= \|\frac{1}{m}\sum_{l=1}^m x_{j_l} - x_{j_1}\|^2.
\end{aligned}
\tag{21}
$$

We observe that $x_k^*$ is a local maximizer of $\min_x d_{\sigma_k}(x;\mathcal{D})^2$, which raises contradiction.

Similarly, we prove that $x_j$ is not unique if $x_k^*$ is a local maximizer or saddle point by contradiction. Assume $x_j$ is the unique minimizer of $\min_{x_i \in \mathcal{D}} \|x_k^* - x_i\|^2$, we have

$$
\begin{aligned}
\lim_{k\to\infty} d_{\sigma_k}(x_k^*;\mathcal{D})^2 &= \min_i \lim_{k\to\infty} \|x_k^* - x_i\|^2 \\
&= \lim_{k\to\infty} \|x_k^* - x_j\|^2 \\
&= 0.
\end{aligned}
\tag{22}
$$

Hence, $x_k^*$ can not be a local maximizer or saddle point of $\min_x d_{\sigma_k}(x;\mathcal{D})^2$, which raises contradiction. $\square$

Since gradient descent converges to a local minimizer almost surely with random initialization [74] and appropriate step sizes, there is a unique $x_j$ for $x_k^*$ and (18) holds. We also note that a similar conclusion can be reached by using $\Gamma$-convergence of the softmin to min as $\sigma_k \to 0$ [75].

## A.2 Proof for Proposition 2

Let $L_e$ be the local Lipschitz constant of the error $e(x) := f(x) - f_\theta(x)$ over some domain $\mathcal{Z} \subseteq \mathcal{X}$, and define $x_c := \arg\min_{x_i \in \mathcal{D}} \frac{1}{2}\|x - x_i\|_2^2$, i.e., the closest data-point. Then, we have the following:

$$
\begin{aligned}
\|f(x) - f_\theta(x)\| &\leq \min_{x_i \in \mathcal{D}} \left[ e(x_i) + L_e \|x - x_i\|_2 \right] \\
&\leq e(x_c) + L_e \|x - x_c\|_2 \\
&= e(x_c) + \sqrt{2} L_e \sqrt{\frac{1}{2}\|x - x_c\|_2^2} \\
&= e(x_c) + \sqrt{2} L_e \sqrt{\frac{1}{2} \min_{x_i \in \mathcal{D}} \|x - x_i\|_2^2} \\
&= e(x_c) + \sqrt{2} L_e \sqrt{\min_{x_i \in \mathcal{D}} \frac{1}{2}\|x - x_i\|_2^2} \\
&\leq e(x_c) + \sqrt{2} L_e \sqrt{-\sigma^2 \log\left( \sum_i \exp\left( -\frac{1}{2\sigma^2}\|x - x_i\|^2 \right) \right) + \sigma^2 \log N} \\
&= e(x_c) + \sqrt{2} L_e \sqrt{d_\sigma(x; \mathcal{D})^2 + C_2}
\end{aligned}
\tag{23}
$$

where $C_2 := \sigma^2 \log N - C$, and $C$ is the constant defined in (4). In the second line, we use the fact that as $x_c$ is a feasible solution to the minimization in the first line, it is an upper bound on the optimal value. In the sixth line, we have used the fact that for any vector $v = [v_1, \ldots, v_n]^\top \in \mathbb{R}^n$, $\min\{v_1, \ldots, v_n\} \leq -\frac{1}{t} \log \sum_{i=1}^n \exp(-tv_i) + \frac{\log n}{t}$ for some scaling $t$. In the final line, we have applied the definition of $d_\sigma(x; \mathcal{D})$ from (4).

## A.3 Proof for Proposition 3

The perturbed data distribution can be written as a sum of Gaussians, since

$$
\begin{aligned}
p_\sigma(x') &:= \int \hat{p}(x) \mathcal{N}(x'; x, \sigma^2 \mathbf{I}) dx \\
&= \int \left[ \frac{1}{N} \sum_i \delta(x_i) \right] \mathcal{N}(x'; x, \sigma^2 \mathbf{I}) dx \\
&= \frac{1}{N} \sum_i \int \delta(x_i) \mathcal{N}(x'; x, \sigma^2 \mathbf{I}) \\
&= \frac{1}{N} \sum_i \mathcal{N}(x_i; x, \sigma^2 \mathbf{I}) \\
&= \frac{1}{N} \sum_i \mathcal{N}(x; x_i, \sigma^2 \mathbf{I})
\end{aligned}
\tag{24}
$$

Then we consider the negative log of the perturbed data distribution multiplied by $\sigma^2$,

$$
\begin{aligned}
-\sigma^2 \log p_\sigma(x) &= -\sigma^2 \log\left[\frac{1}{N}\sum_i \mathcal{N}(x; x_i, \sigma^2 \mathbf{I})\right] \\
&= -\sigma^2 \log\left[\sum_i \mathcal{N}(x; x_i, \sigma^2 \mathbf{I})\right] + \log N \\
&= -\sigma^2 \log\left[\frac{1}{\sqrt{(2\pi\sigma)^n}}\sum_i \exp\left[-\frac{1}{2\sigma^2}\|x - x_i\|^2\right]\right] + \sigma^2 \log N \\
&= -\sigma^2 \log\left[\sum_i \exp\left[-\frac{1}{2\sigma^2}\|x - x_i\|^2\right]\right] + \sigma^2 \log N + \sigma^2 \frac{n}{2}\log(2\pi\sigma) \\
&= -\sigma^2 \mathsf{LogSumExp}_i\left[-\frac{1}{2\sigma^2}\|x - x_i\|^2\right] + C(N, n, \mathbf{\Sigma}) \\
&= \mathsf{Softmin}_\sigma\left[\frac{1}{2}\|x - x_i\|^2\right] + C(N, n, \mathbf{\Sigma})
\end{aligned}
\tag{25}
$$

where we define $C(N, n, \mathbf{\Sigma}) := \sigma^2(\log N + n/2\log(2\pi\sigma))$.

## A.4   Function Error vs. Gradient Error

We illustrate with an example that in the finite error regime, bounded function error does not necessarily imply bounded gradient error.

Suppose we are given a function $f(x) : \mathbb{R} \to \mathbb{R}$, and another function of the form

$$
g(x) = f(x) + \alpha \cos(\omega x)
\tag{26}
$$

Then, the error between the two functions is bounded by

$$
e(x) := \|f(x) - g(x)\| \le \alpha \cos(\omega x) \le \alpha
\tag{27}
$$

for all $x \in \mathbb{R}$. One might make $\alpha$ arbitrarily small (but not zero) to decrease the error.

However, consider the error in the gradients,

$$
e_\nabla(x) := \|\nabla f(x) - \nabla g(x)\| \le \alpha\omega \sin(\omega x) \le \alpha\omega
\tag{28}
$$

which now scales with the frequency term $\omega$, which can be arbitrarily scaled up to increase gradient error.

# B   Details of the Planning algorithm

## B.1   Computation of Gradients

Recall that the gradient of $\log p_\sigma(x_i, u_i)$ cost w.r.t. the input variable $u_j$ can be written as

$$
\nabla_{u_j} \log p(x_i, u_i) = \nabla_{x_i} \log p(x_i, u_i)\mathbf{D}_{u_j} x_i + \nabla_{u_i} \log p(x_i, u_i)\mathbf{D}_{u_j} u_i
\tag{29}
$$

where $\mathbf{D}$ denotes the Jacobian. Writing the dependence on each variable more explicitly, we have

$$
\nabla_{u_j} \log p(x_i(u_j), u_i(u_j)) = \nabla_{x_i} \log p(x_i(u_j), u_i(u_j))\mathbf{D}_{u_j} x_i(u_j)
\tag{30}
$$
$$
+ \nabla_{u_i} \log p(x_i(u_j), u_i(u_j))\mathbf{D}_{u_j} u_i(u_j)
\tag{31}
$$

where we note that $\mathbf{D}_{u_j} u_i = 1$ if $i = j$ and 0 otherwise. As long as $i > j$, we also note that $x_i$ has a dependence on $u_j$. Instead of computing this gradient explicitly, we first rollout the trajectory to compute $x_i(u_j), u_i(u_j)$, and compute the score function. Then we ask: which quantity do we need such that it gives us the above expression when differentiated w.r.t. $u_j$? We use the following quantity,

$$
c_{ij} = s_x(x_i, u_i)x_i(u_j) + s_u(x_i, u_i)u_i(u_j)
\tag{32}
$$

where the score terms have been detached from the computation graph. Note that $c_{ij}$ is a scalar and allows us to use reverse-mode automatic differentiation tools such as `pytorch` [76].

## B.2   Noise-Annealing During Optimization

Additionally, we anneal the noise level during iterations of Adam. Given a sequence $\sigma_k$ with $K$ being the total number of annealing steps, we run Adam for max $\text{iter}_{\max}/K$ iterations, then run it with the next noise level.

## B.3   Algorithm Block

We first give some abbreviations to simply the algorithm description. Given the dynamics, we write down the value function of Equation (11), conditioned in the noise level $\sigma$, as

$$V(u_{1:T}; \sigma) := \sum_{t=1}^{T} r_t(x_t, u_t) + \beta\sigma^2 \sum_{t=1}^{T} \log p_\sigma(x_t, u_t; \mathcal{D}) \tag{33}$$

$$\text{s.t.}\quad x_{t+1} = f_\theta(x_t, u_t) \quad \forall t$$

We note that in our implementation, we use Adam [50] instead of doing gradient descent.

---

**Algorithm 1** Score-Guided Planning (Gradient Descent Version)

---

**Require:** Learned dynamics $f_\theta$, score function $s(x, u)$, noise schedule sequence $\sigma_k$.
**Require:** Initial guess $u_{1:T}^0$, initial noise $\sigma_0$.
  **while  not** converged **do**
      Rollout $u_{1:T}^k$ and compute state-input trajectory $(x_t, u_t)^k$.
      Compute $\nabla_{u_{1:T}} V(u_{1:T}^k; \sigma_k)$ using score estimator $s(x, u, \sigma)$  ▷ Equation (12), Appendix B.1
      $u_{1:T}^{k+1} \leftarrow u_{1:T}^k - h\nabla_{u_{1:T}} V(u_{1:T}^k, \sigma_k)$           ▷ Gradient Descent with stepsize $h$
      $k \leftarrow k + 1$
  **end while**

---

## B.4   Connection to Diffuser

We first lift the dynamics constraint into a quadratic penalty and write the penalty as $\log p(x_{t+1}|x_t, u_t)$. This equivalence is seen by considering a case where we fix $x_t, u_t$ and perturb $x_{t+1}$ with a Gaussian noise of scale $\sigma$. If $(x_t, u_t, x_{t+1})$ is in the dataset, it obeys $x_{t+1} = f(x_t, u_t)$ under real-world dynamics $f$. This allows us to write

$$p_\sigma(x_{t+1}|x_t, u_t) = \mathcal{N}(x_{t+1}|f(x_t, u_t), \sigma^2\mathbf{I})$$

$$\log p_\sigma(x_{t+1}|x_t, u_t) = -\frac{1}{2}\|x_{t+1} - f(x_t, u_t)\|^2 + C \tag{34}$$

where $C$ is some constant that does not effect the objective. Then, we rewrite our objective using the factoring $p(x_t, u_t) = p(u_t|x)p(x_t)$. This allows us to rewrite the objective of Equation (11) as

$$\sum_{t=1}^{T} r_t(x_t, u_t) + \beta\sum_{t=1}^{T}\log p(u_t|x_t) + \beta\sum_{t=1}^{T}\log p(x_t) + \beta\sum_{t=1}^{T}\log p(x_{t+1}|x_t, u_t)$$

$$= V(x_{1:T}, u_{1:T}) + \beta\log p(x_{1:T}, u_{1:T}) + \sum_{t=1}^{T}\log p(x), \tag{35}$$

where the first two terms are the objectives in Diffuser [26].

## B.5   First-Order Policy Search

We note that our original method for gradient computation can easily be extended to the setting of feedback first-order policy search, where we define the uncertainty-penalized value function as

$$\max_{\alpha}\quad \mathbb{E}_{x_1\sim\rho}\left[\sum_{t=1}^{T} r_t(x_t, u_t) + \beta\sigma^2\sum_{t=1}^{T}\log p_\sigma(x_t, u_t; \mathcal{D})\right] \tag{36}$$

$$\text{s.t.}\quad x_{t+1} = f_\theta(x_t, u_t), u_t = \pi_\alpha(x_t) \quad \forall t \in [1, T],$$

where $\rho$ is some distribution of initial conditions. We rewrite the objective with an explicit dependence on $\alpha$, and use a Monte-Carlo estimator for the gradient of the stochastic objective,

$$\nabla_\alpha \mathbb{E}_{x_1 \sim \rho} \left[ \sum_{t=1}^T r_t(x_t(\alpha), u_t(\alpha)) + \beta \sigma^2 \sum_{t=1}^T \log p_\sigma(x_t(\alpha), u_t(\alpha); \mathcal{D}) \right]$$

$$= \mathbb{E}_{x_1 \sim \rho} \nabla_\alpha \left[ \sum_{t=1}^T r_t(x_t(\alpha), u_t(\alpha)) + \beta \sigma^2 \sum_{t=1}^T \log p_\sigma(x_t(\alpha), u_t(\alpha); \mathcal{D}) \right]$$

$$\approx \frac{1}{N} \sum_{i=1}^N \nabla_\alpha \left[ \sum_{t=1}^T r_t(x_t(\alpha), u_t(\alpha)) + \beta \sigma^2 \sum_{t=1}^T \log p_\sigma(x_t(\alpha), u_t(\alpha); \mathcal{D}) \quad \text{s.t.} \quad x_1 = x_i \sim \rho \right],$$

$$(37)$$

where the last equation denotes that fixing the initial condition to $x_i$ sampled from $\rho$, and $N$ is the number of samples in the Monte-Carlo process. Since $r_t$ and $f_\theta$ are differentiable, we can obtain the gradient

$$\nabla_\alpha \sum_{t=0}^T r_t(x_t(\alpha), u_t(\alpha)) \tag{38}$$

after rolling out the closed-loop system starting from $x_i$ and using automatic differentiation w.r.t. policy parameters $\alpha$. To compute the gradient w.r.t. the score function, we similarly use the chain rule,

and differentiate it w.r.t $\alpha$, which lets us compute

$$\frac{\partial}{\partial \alpha} \left[ \sum_{t=1}^T \log p_\sigma(x_t, u_t) \right] = \sum_{t=1}^T \frac{\partial}{\partial \alpha} \log p_\sigma(x_t(\alpha), u_t(\alpha))$$

$$= \sum_{t=1}^T \frac{\partial}{\partial x_t} \log p_\sigma(x_t, u_t) \frac{\partial x_t}{\partial \alpha} + \frac{\partial}{\partial u_t} \log p_\sigma(x_t, u_t) \frac{\partial u_t}{\partial \alpha} \tag{39}$$

$$= \sum_{t=1}^T s_x(x_t, u_t; \sigma) \frac{\partial x_t}{\partial \alpha} + s_u(x_t, u_t; \sigma) \frac{\partial u_t}{\partial \alpha}$$

where the last term is obtained by differentiating

$$\sum_{t=1}^T s_x(x_t, u_t; \sigma) x_t + s_u(x_t, u_t; \sigma) u_t \tag{40}$$

after detaching $s_x$ and $s_u$ from the computation graph.

## B.6 Imitation Learning

We give more intuition for why maximizing the state-action likelihood leads to imitation learning. If the empirical data comes from an expert demonstrator, maximizing data likelihood leads to minimization of cross entropy between the state-action pairs encountered during planning and the state-action occupation measure of the demonstration policy, which is estimated with the perturbed empirical distribution $p_\sigma(x_t, u_t)$,

$$\sum_t \log p_\sigma(x_t, u_t) = \sum_t \log p_\sigma(u_t | x_t) + \sum_t \log p_\sigma(x). \tag{41}$$

Note that the $\log p_\sigma(u_t | x_t)$ is identical to the Behavior Cloning (BC) objective, while $\log p_\sigma(x_t)$ drives future states of the plan closer to states in the dataset. We note that Adversarial Inverse Reinforcement Learning (AIRL) [35] minimizes a similar objective as ours [13].

# C Experiment Details

## C.1 Cartpole with Learned Dynamics

**Environment.** We use the cart-pole dynamics model in [62, chapter 3.2], with the cost function being

$$c_t(x_t, u_t) = \begin{cases} \|x_t - x_g\|_{\mathbf{Q}}^2 & \text{if } t = T \\ 0 & \text{else.} \end{cases} \qquad (42)$$

$\mathbf{Q} = \text{diag}(1, 1, 0.1, 0.1)$. We choose the planning horizon $T = 60$.

**Training.** We randomly collected a dataset of size $N = 1,000,000$ within the red box region in the state space. The dynamics model is an MLP with 3 hidden layers of width $(64, 64, 32)$. For ensemble approach we use 6 different dynamics models, all with the same network structures.

We train a score function estimator, represented by an MLP with 4 hidden layers of width 1024. The network is trained form 400 epochs with a batch size of 2048.

**Parameters** During motion planning, for Adam optimizer we use a learning rate of $0.01$. For CEM approach we use a population size of 10, with standard variance $\sigma = 0.05$, and we take the top 4 seeds to update the mean in the next iteration.

**Data Distance Estimator.** To train a data distance estimator, we introduce a function approximator $d_\eta : \mathbb{R}^n \times \mathbb{R}_+ \to \mathbb{R}$ parametrized by $\eta$ to predict the noise-dependent Softmin distance. The training objective is given by

$$\min_\eta \frac{1}{2} \mathbb{E}_{x \in \Omega, \sigma \in [0, \sigma_{max}]} \left[ \frac{1}{\sigma^2} \left| d(x, \sigma) - \text{Softmin}_{x_i \in \mathcal{D}} \frac{1}{2} \|x - x_i\|_{\sigma^{-2}\mathbf{I}}^2 \right| \right] \qquad (43)$$

where $\Omega$ is a large enough domain that covers the data distribution $\mathcal{D}$. For small datasets, it is possible to loop through all $x_i$ in the dataset to compute this loss at every iteration. However, this training can get prohibitive as all of the training set needs to be considered to compute the loss, preventing batch training out of the training set.

## C.2 D4RL Dataset

**Environment.** We directly use the D4RL dataset [44] Mujoco tasks [64] with 3 different environments of halfcheetah, walker2d, and hopper. We additionally use differnet sources of data with random, medium, and medium-expert.

**Training.** The dynamics and the score functions are both parametrized with MLP with 4 hidden layers of width 1024. The noise-conditioned score function is implemented by treating each level of noise $\sigma_k$ as an integer token, that gets embedded into a 1024 vector and gets multiplied with the output of each layer. This acts similar to a masking of the weights depending on the level of noise. The D4RL environment does not provide us with a differentiable reward function, so we additionally train an estimator for the reward. We empirically saw that for score function estimation, wide shallow networks performed better. We train both instances for 1000 iterations with Adam, with a learning rate of $1e-3$ and batch size of 2048.

We additionally set a noise schedule to be a cosine schedule that anneals from $\sigma = 0.2$ to $\sigma = 0.01$ for 10 steps in the normalized space of $x, u$.

**Parameters** We used a range of $\beta$s between $1e^{-3}$ and $1e^{-1}$ depending on the environment, where in some cases it helped to be more reliant on reward, and in others it's desirable to rely on imitation. We use a MPC with $T = 5$ and optimize it for 50 iterations with an aggressive learning rate of $1e^{-1}$.

## C.3 Pixel Single Integrator

**Environment.** In this environment, we have a 2D single integrator $f(x_t, u_t) = x_t + u_t$, $x_t \in \mathbb{R}^2$, $u_t \in \mathbb{R}^2$ as the underlying ground-truth dynamics; however, instead of raw states $x_t$, we observe a $32 \times 32$ grayscale image $y_t = h(x_t) \in \mathbb{R}^{32 \times 32}$, which are top-down renderings of the robot. In these observations, the position of the robot is represented with a dot. Moreover, we assume that we do not directly assign the 2D control input $u_t$, but instead propose a $32 \times 32$ grayscale image $\hat{u}_t$, where the value of the 2D control action $u_t$ is extracted from the image via a spatial average:

$$u_t = \sum_{(p_x, p_y)} g_u(p_x, p_y) \hat{u}_t(p_x, p_y), \tag{44}$$

where the sum loops over each pixel $(p_x, p_y) \in \{1, \ldots, 32\}^2$, $y_t(p_x, p_y)$ refers to the intensity of the image at pixel $(p_x, p_y)$, and $g_u : \mathbb{Z} \times \mathbb{Z} \to \mathbb{R}^2$ is a grid function mapping from pixel $(p_x, p_y)$ to a corresponding control action. The control image $\hat{u}_t$ is normalized such that its overall intensity sums to 1. Given some goal $x_g$, the reward is set to be the $-c_t(x_t, u_t)$, where the cost $c_t(x_t, u_t)$ is

$$c_t(x_t, u_t) = \begin{cases} \|x_t - x_g\|^2_{\mathbf{Q}_t} + \|u_t\|^2_{\mathbf{R}} & \text{if } t = T \\ \|x_t - x_g\|^2_{\mathbf{Q}} + \|u_t\|^2_{\mathbf{R}} & \text{else,} \end{cases} \tag{45}$$

To evaluate this cost function for the planned sequence of image observations and control images, the states $x_t$ are also extracted from the image observations through a similar spatial averaging:

$$x_t = \sum_{(p_x, p_y)} g_{\text{im}}(p_x, p_y) y_t(p_x, p_y), \tag{46}$$

where $g_{\text{im}} : \mathcal{Z} \times \mathcal{Z} \to \mathbb{R}^2$ is a grid function mapping from pixel $(p_x, p_y)$ to a corresponding state.

In other words, we have running costs for the state and input, and a different terminal cost for the state. We set $\mathbf{R} = 6.5\mathbf{I}$, $\mathbf{Q} = 500\mathbf{I}$, and $\mathbf{Q}_d = 1000\mathbf{I}$, and plan with a horizon of $T = 15$.

**Training.** We collect a randomly collected dataset of size $N = 200,000$, with underlying 2D data sampled from $x_t \in [-1, 1]^2$ and $u_t \in [-0.2, 0.2]^2$. Both the dynamics and the score function estimator are represented as U-Nets [77], with the architecture coming from [78].

**Parameters.** We found that $\beta = 0.5$ is sufficient for the penalty parameter. Results are obtained within 1450 iterations with a learning rate of 0.03. In representing the noise-conditioned score function, we use 232 smoothing parameters $\{\sigma_k\}_{k=1}^{232}$, from $\sigma_1 = 50$ to $\sigma_{232} = 0.01$.

**Baselines.** For gradient-based planning with ensembles, we set $\beta = 0.5$ and use an ensemble of size 10. For CEM with ensembles, we set $\beta = 0.5$, with an ensemble of size 5 (we only used the first five networks in the original ensemble of size 10 due to RAM limitations).

| Method | Cost of plan | Actual achieved cost |
|---|---|---|
| SGP (Ours) | 87.61 | **124.72** |
| Vanilla MBRL | 0.80 | 1862.06 |
| Ensembles | 4031.54 | 1380.41 |
| CEM | 2244.72 | 2901.95 |

Table 2: Costs for pixel-space single integrator example (lower is better).

**Numerical results.** In Table 2, we report numerical results on the costs achieved by our approach and the various baselines tested in the pixel-space single integrator example (Fig. 3 in the main text). SGP (our method, Method A in Fig. 3) achieves a low cost at planning time (87), and the actual achieved cost when rolling out the controls open-loop on the true system is only slightly worse (124), since the model error is kept small. In contrast, for vanilla MBRL (Method B in Fig. 3 in the main text), an overly-optimistic cost is achieved (0.8) at planning time, due to the generation of unrealistic images that exploit subtleties in how the cost is calculated; however, when executing the resulting

controls, the cost greatly degrades (1862) due to the model mismatch. Using ensembles (Method C in Fig. 3 in the main text) leads to the optimizer failing to find good control sequence at planning time, due to the poor optimization landscape when planning with ensembles (cost is 4031); this translates to a high cost when rolling out on the true system as well (cost is 1380). CEM (Method D in Fig. 3 in the main text) similarly does not find a good control sequence at planning time, due to the high dimensionality of the search space (cost of 2244); the actual cost achieved is similarly high (cost of 2901).

## C.4 Box Pushing with Marker Dynamics

**Environment.** We prepare a box-pushing environment where we assume that the box follows quasistatic dynamics, which allows us to treat the marker positions directly as state of the box that is bijective with its pose. We use 2D coordinates for each markers, and append the pusher position, also in 2D, resulting in $x_t \in \mathbb{R}^{12}$. The pusher is given a relative position command with a relatively large step size [73]. In addition, we give the robot knowledge of the pusher dynamics, $x_{t+1}^{\text{pusher}} = x_t^{\text{pusher}} + u_t$. The general goal of the task is to push the box and align the edge of the box with the blue tape line.

We formulate our cost as

$$c_t(x_t, u_t) = \begin{cases} \|x_t^{\text{marker}} - x_g^{\text{marker}}\|_{\mathbf{Q}_T}^2 & \text{if } t = T \\ \|u_t\|_{\mathbf{R}}^2 & \text{else,} \end{cases} \tag{47}$$

where $\mathbf{Q}_T = \mathbf{I}$ and $\mathbf{R} = 0.1\mathbf{I}$. In order to get $x_g^{\text{marker}}$, we place the box where we want the goal to be and measure the position of the markers.

**Training.** We collect 100 demonstration trajectories resulting in 750 pairs of $(x_t, u_t, x_{t+1})$. The dynamics and the score functions are learned with a MLP of 4 hidden layers with size 1024, with the noise-conditioned score estimator being trained similar to the D4RL dataset with multiplicative token embeddings. We train for 500 iterations with a batch size of 32.

We additionally set a noise schedule to be a cosine schedule that anneals from $\sigma = 0.2$ to $\sigma = 0.01$ for 10 steps.

**Parameters.** We observed that $\beta = 1e^{-2}$ performs well for all the examples, with a learning rate of 0.1 and 50 iterations. We found that a horizon of $T = 4$ was sufficient for our setup.

**Numerical Results** We report numerical results for the box pushing demo in Table C.4, which illustrates that SGP achieves muchy lower cost compared to vanilla MBRL.

| Goal | Method | Achieved Cost |
|--------|-------------|---------------|
| Left | SGP (Ours) | **8.86** $\pm$ 4.54 |
| Left | Vanilla MBRL | 51.18 $\pm$ 6.93 |
| Center | SGP (Ours) | **7.74** $\pm$ 2.23 |
| Center | Vanilla MBRL | 35.93 $\pm$ 1.18 |
| Right | SGP (Ours) | **9.05** $\pm$ 3.45 |
| Right | Vanilla MBRL | 61.40 $\pm$ 3.82 |

Table 3: Cost for the Keypoints hardware Example (lower is better). We compare the results of running MPC using vanilla MBRL, vs. SGP. The cost is evaluated on the final achieved trajectory on hardware. Numbers are averaged on 5 trials.

# D Experiments on Hyperparameters on SGP

## D.1 Effect of Penalty term $\beta$.

Note that $\beta$ appears in the objective to trade off the reward signal and the uncertainty penalty term,

$$\max_{x_{1:T}, u_{1:T}} \quad \sum_{t=1}^{T} r_t(x_t, u_t) + \beta \sigma^2 \sum_{t=1}^{T} \log p_\sigma(x_t, u_t; \mathcal{D}) \tag{48}$$
$$\text{s.t.} \quad x_{t+1} = f_\theta(x_t, u_t) \quad \forall t \in [1, T].$$

We first give intuition for two extreme cases.

### D.1.1 Vanilla Model-Based Reinforcement Learning (MBRL), $\beta = 0$

When $\beta$ is 0, no uncertainty is penalized, and the planning problem becomes equivalent to vanilla Model-Based Reinforcement Learning (MBRL), or planning with learned dynamics. While this achieves best performance in terms of $f_\theta$, the optimizer is likely to not perform well under the true dynamics $f$ due to model bias.

### D.1.2 Imitation Learning / Data Landing, $\beta = \infty$

When $\beta = \infty$, the reward term is discarded and the problem simply turns to that of uncertainty minimization. When data comes from random sources, the problem will then plan a trajectory to stay near the vicinity of the data depending on the value of $\sigma$.

Interestingly, the case for when $\beta = \infty$ can be understood as a case of imitation learning if data comes from an expert demonstrator. We can write down

$$\sum_{t=1}^{T} \log p_\sigma(x_t, u_t; \mathcal{D}) = \sum_{t=1}^{T} \mathsf{KL}(\rho(x_t, u_t) \| \tilde{p}(x_t, u_t)) \tag{49}$$

where $\rho$ is the occupancy measure of the plan, and thus maximizing this quantity leads to a form of distribution matching between this occupancy measure and the perturbed empirical distribution of data. We have more details in Appendix B.6.

### D.1.3 Experimental Validation

To illustrate the effects of $\beta$, we prepare a single-integrator environment where the dynamics obey $x_{t+1} = x_t + u_t$, except in the circular region where actuation gets lost and $x_{t+1} = x_t$. For instance, this could simulate a pit in a self-driving environment. The effect of setting different $\beta$s are illustrated in Fig.5.

As hypothesized, too high of a $\beta$ results in not making progress due to no reward signal - as a result, the cost is quite high. Yet, the dynamics error is very low. On the other hand, too low of a $\beta$ results in a big dynamics error as the single integrator goes into the middle pit region and makes no progress. As a result, the overall cost is very high as well.

We note that around $\beta = 1e^{-2}$ and $\beta = 1e^{0}$ is the sweet spot for this experiment, where the agent was able to circumnavigate the pit as it has not seen data there, and still make meaningful progress towards the goal.

### D.1.4 Tuning Recommendation

In practice we recommend setting $\beta$ between values of $\beta = 1e^{-3}$ to $\beta = 1e^{0}$ and explore performance tradeoffs by doing hyperparameter search.

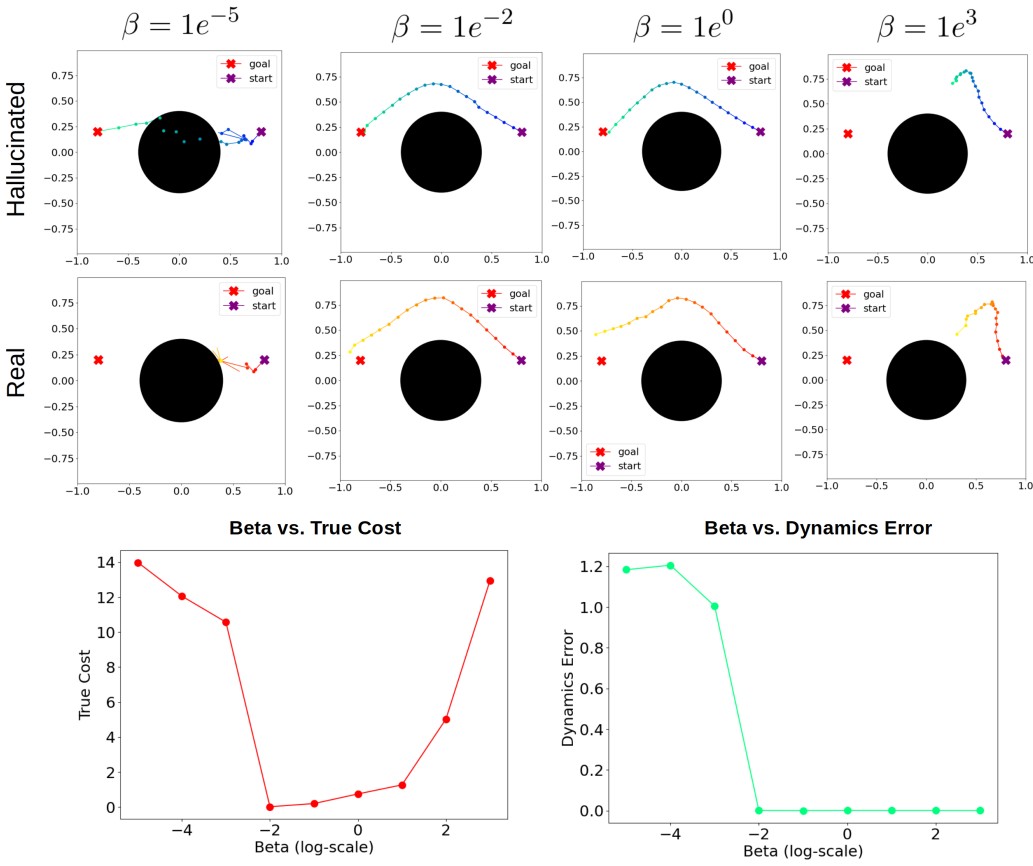

Figure 5: Top row: The hallucinated trajectory with different choices of $\beta$. Middle row: real rollouts with different choices of $\beta$. Bottom row, left: sweep of $\beta$ in log-scale and cost. Bottom row, right: sweep of $\beta$ and the dynamics error on the rolled out trajectory.

## D.2 Effect of Noise Level $\sigma$.

Similar to $\beta$, the performance of the algorithm deteriorates under too high / too low of a value for the injected noise $\sigma$.

### D.2.1 Effect of low $\sigma$

The biggest factor preventing us from having too low of a $\sigma$ comes from the learning of the score function. As denoising score matching works by perturbing the data with the given noise $\sigma$, only regions within a small vicinity of the data will have been covered by score matching if $\sigma$ is low. As the training accuracy of neural networks deteriorates on regions it has not been trained on, the accuracy of score matching suffers greatly with too low of a $\sigma$. This effect is extensively illustrated by previous score matching papers such as [40].

Another effect is that without normalization by $\sigma$, the log probabilities of the perturbed empirical distribution has a scaling that directly depends on $\sigma$. This is clear to see with a single data, for which the perturbed empirical distribution becomes a Gaussian.

$$-\log p \approx \frac{x^2}{2\sigma^2} + C \tag{50}$$

This suggests that as $\sigma$ becomes too small, the Lipschitz constant of the distribution becomes large, creating a landscape that is challenging for neural networks to learn.

To visually validate this, we illustrate the efficacy of score matching as we vary $\sigma$ in Fig.6. As we hypothesize, score matching is very ineffective for small $\sigma$, and the actual gradients are very jagged / near-discontinuous in this regime. On the other hand, as we increase $\sigma$ to around $0.1$ and $0.2$ level, score matching successfully predicts gradients of log-likelihood. However, even for $\sigma = 0.2$, note that we cannot predict the score accurately if we get too far away from the data, since the score estimator never sees these points during training.

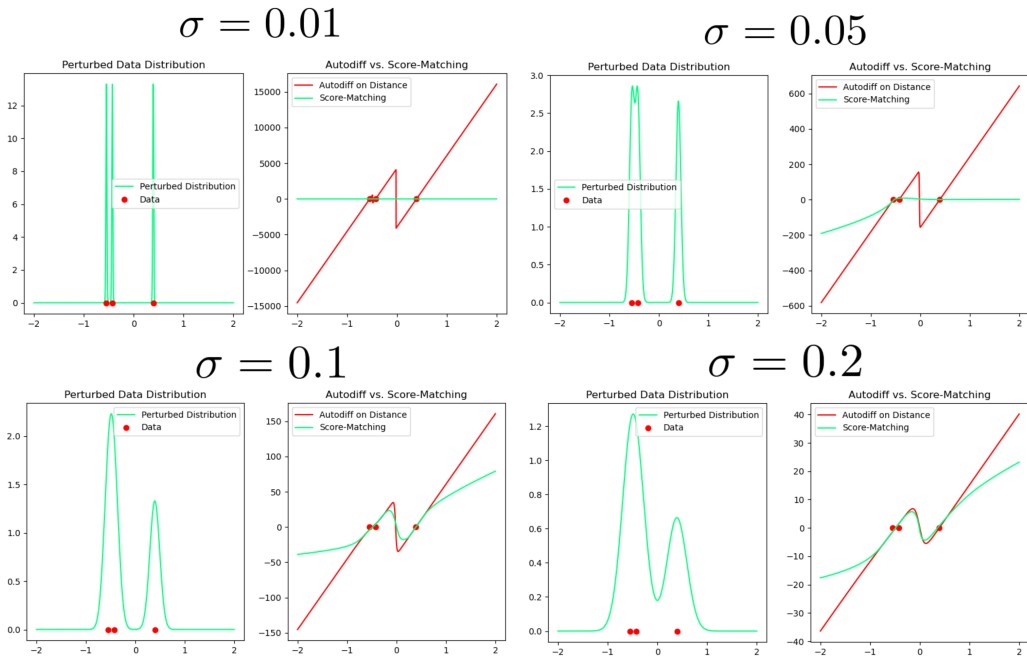

Figure 6: Efficacy of Score Matching as we vary the perturbed noise $\sigma$. note that this $\sigma$ is in the unnormalized space.

### D.2.2 Effect of high $\sigma$

If $\sigma$ is too high compared to the scale of how close data are together, then it is hard to distinguish from a pure Gaussian noise centered around the average of the data. Indeed, this is what the "nosing process" of diffusion models rely on.

As the goal of SGP is to stay near the data, not go towards the average of data, it is not desirable to choose too high of a $\sigma$. In fact, we provide an adversarial case in the pit example, where we have not collected any data near the mean of the data. In such cases, too high of a $\sigma$ can lead to a catastrophic failure.

### D.2.3 Experimental Validation

We validate our choice of $\sigma$ by training a score estimator for the example in Appendix D.1.3 and visualizing its behavior. Our experiments validate our hypothesis that when $\sigma$ is too small, the score function is very inaccurate. On the other hand, too big of a $\sigma$ leads the score function to point towards the center of all the data - which in this case, corresponds to a region where data was not collected. In between we have a sweet spot where the score function pushes away from regions with no data.

### D.2.4 Tuning Recommendation

To see whether a user has chosen a good $\sigma$, we recommend that the user first tests the validity of the score estimator before attempting to solve any optimal control problem. This can be done by training a score estimator, and running gradient descent on the log probabilities using the score estimator as

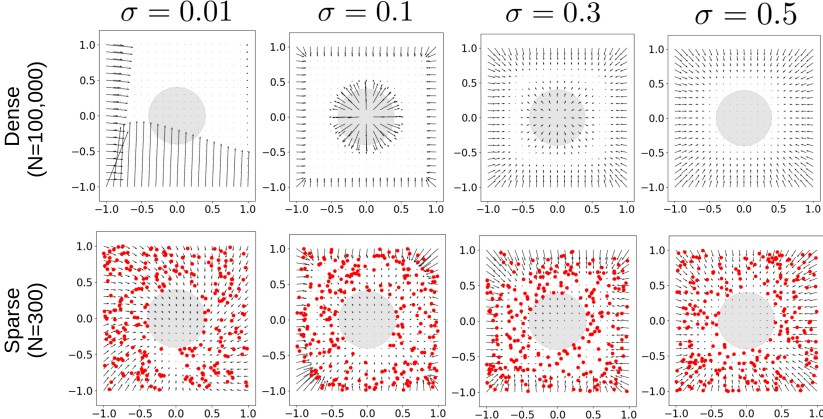

Figure 7: Visualization of the learned score function $s(x) = \nabla_x \log \tilde{p}_\sigma(x)$, according to different $\sigma$ values. Top corresponds to a dense and bottom corresponds to a sparse dataset. The grey region corresponds to region with no data. Note that $\sigma$ corresponds to noise in the normalized space.

done in [40]. A good score estimator should ensure that after some iterations, the iterates converge to samples within the data.

We note that in the normalized space, we often find $\sigma = 0.05 \sim 0.3$ to be good. To tradeoff between accuracy and coverage within this range, we anneal $\sigma$ in between this range following [40].

# E  Experiments on Extrapolation

One limitation of SGP that was mentioned was that the method does not have the ability to extrapolate. We show that,

1. On the trajectory level, our method can successfully extrapolate by stitching trajectories to create a trajectory that was not in the dataset.
2. Not doing extrapolation can be a very practical method on pathological cases where extrapolation actually hurts.

## E.1  Trajectory-level Extrapolation by Stitching

In order to show our trajectory-level extrapolation capabilities, we replicate the example in [26] by preparing two types of demonstration trajectories. The first type of demonstration goes from the bottom left to right top, while the second type of demonstration goes from top left to bottom right. Then, SGP is asked to create a trajectory that goes from bottom left to bottom right - as this kind of demonstration has never been seen before, this requires use to extrapolate in the space of trajectories and perform a type of trajectory stitching.

As our results in Figure 8 indicates, SGP is successful in stitching sub-trajectories from the two demonstrations to come up with an entirely new trajectory that was not in the demonstration set. Thus, our method is capable of performing extrapolation in the space of trajectories.

## E.2  Dynamics-Level Extrapolation as a double-edged sword

By actively constraining the state-action pairs to stay close to the dataset, SGP discourages dynamics-level extrapolation. While this can be a possible limitation, we also show that this is an important strength of our method compared to methods that attempt to extrapolate.

For this purpose, we consider the single-integrator with a data-hole inside, where the agent has not seen data before. The true dynamics is defined as

$$x_{t+1} = \begin{cases} x_t + u_t & \text{if outside hole} \\ x_t & \text{if inside hole} \end{cases} \tag{51}$$

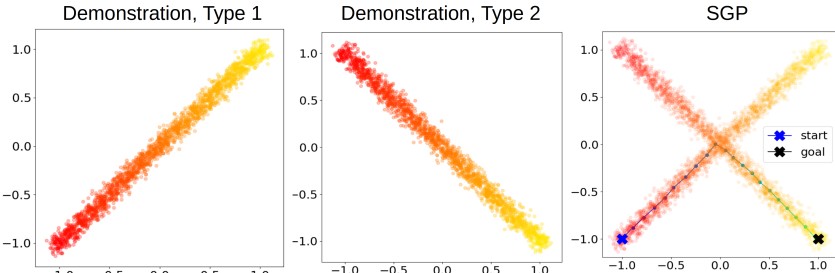

Figure 8: Left: Demonstration trajectories that go from left bottom to right top. Center: demonstration trajectories that go from left top to right bottom. Right: performance of SGP when asked to find a trajectory from left bottom to right bottom, which was not in the dataset before.

These scenarios are plausible if the hole region falls outside the normal operating conditions of the agent. For example, a self-driving car might always be forced to go around a pit, or patches of grass in a rotary. Naturally the dynamics are still defined in the physical world, but the agent has never seen data there. We test the performance of SGP vs. Ensembles in this scenario and plot the results in Figure 9

As the dynamics are very simple outside the hole, ensembles will learn the simple dynamics $x_{t+1} = x_t + u_t$ with very high-confidence. As a result, the trajectory found by penalizing ensemble variance attempts to extrapolate, cutting across the data hole region where it believes the dynamics is still a single integrator. Yet; the extrapolation fails and the method gets stuck at making progress.

However, SGP avoids this problem by making no assumptions about the extrapolation region, and simply avoiding it. This allows it to escape the pit region and still succeed in reaching the goal.

This experiment goes to show that attempting extrapolation is a double-edged sword. While the agent may have learned a generalizable form of dynamics that allows it to perform well, it also may not have, and there is no data to test it. Thus, in cases where we do not have enough domain knowledge or inductive bias in the network, it can be a strength to avoid extrapolation rather than limitation.

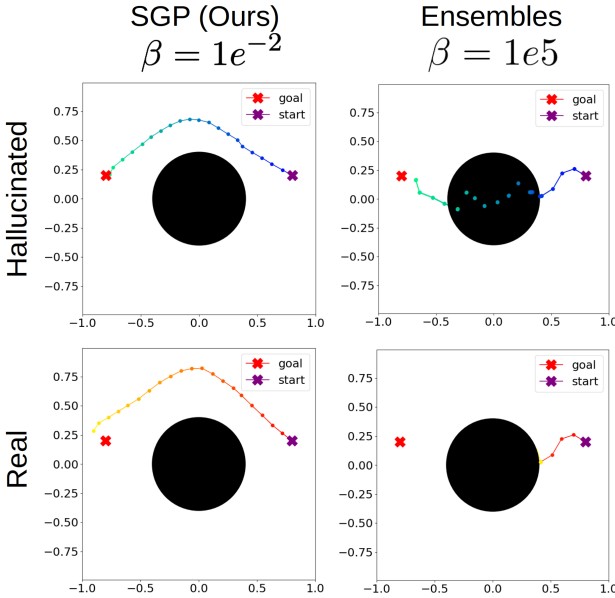

Figure 9: Comparison of SGP vs. Ensembles variance penalty in the single integrator with data hole example. While SGP successfully avoids the hole, ensembles overconfidently extrapolate and fail to plan a good trajectory that transfers to real-world dynamics.

