# OpenReview forum: "Fighting Uncertainty with Gradients: Offline Reinforcement Learning via Diffusion Score Matching"
_robot-learning.org/CoRL/2023/Conference — CoRL 2023 Poster_

### Official Review · Reviewer_9mZ5 · 2023-07-19

**Confidence:** 4
**Originality:** Very Good
**Technical Quality:** Very Good
**Clarity Of Presentation:** Excellent
**Impact:** 4

**Recommendation:**

Weak Accept: I recommend accepting the paper, but will not argue for my recommendation if the majority of other reviewers have a different opinion.

**Review:**

The reinterpretation of score-matching for a perturbed empirical distribution as learning gradients of smoothed distance to data is a clever idea, and the paper is well-written with convincing experiments supporting the authors' main claims. I am pleased to recommend its acceptance, however I believe that there are two areas of improvement which, if addressed, would significantly improve the impact of this paper.

1. *Relevance of the theory:* Section 3 argues that smoothed distance to data is a good choice for gradient-based optimization as it can be guaranteed that, when solely optimizing for this metric, gradient descent will converge to a point in the dataset as the smoothing parameter is taken to 0. This fact is not particularly surprising (e.g., an alternate/more constructive proof could reason about Voronoi cells, minus a shrinking boundary as $\sigma \rightarrow 0$, as regions of attraction), but perhaps critically it overlooks the possibility that selecting a spot between data points may be desirable. A common rationale to use neural networks to fit dynamics is, if not to generalize/extrapolate (e.g., through meaningful learned representations as mentioned in the discussion of limitations), at least to interpolate. In light of this, $\sigma$ may be interpreted not only as a penalty for straying outside of the training dataset, but also as a factor allowing for interpolation. That is, $\sigma$ seems like one of the more critical tuning parameters of this method (similar to how the choice of a kernel bandwidth parameter is regarded as a key tuning parameter for Gaussian process-based methods), and may deserve more empirical study than the theory might suggest.

2. *Experimental evaluation:* As noted above, the experiments in this paper do convincingly demonstrate the applicability of smoothed distance to data to gradient-based trajectory optimization. Beyond that, however, the experiments do not yield much insight on how SGP's performance depends on the choice of its tuning parameters (i.e., $\sigma$ and $\beta$). For example, in line with the previous point, studying sparser data (i.e., lower $N$) or finer time discretization may be a way to study the role of $\sigma$ in allowing for interpolation. Also, arguably all of the experiments are conducted on effectively low-dimensional state/action spaces (by construction in the case of the Pixel-Based Single Integrator). What happens if you try the real-world box-pushing task with image observations instead of mocap marker positions? Though undoubtedly mechanically SGP can perform gradient-based optimization in multi-thousand-dimensional spaces, if the method doesn't scale to tasks where you cannot ensure that a solution trajectory can be assembled as a sequence of states you've seen in the training dataset (i.e., *data stability* is impractical, which would at face value seem to be the case for increasingly high dimensional systems), it may be worth mentioning in the limitations section (e.g., this may be less of an issue for uncertainty quantification methods that attempt to infer model confidence irrespective of distance to data in the input space).

Minor comments:
- In the problem definition of offline model-based RL which mentions an empirical distribution over state transitions (line 93), it initially appears that the dynamics themselves may be stochastic (i.e., contain aleatoric uncertainty). It is mentioned only later in the last line of the limitations that aleatoric uncertainty is uninvestigated; for reader clarity it may be worth mentioning earlier (i.e., in Sections 1 and 2) that the dynamics are assumed deterministic and that learning $f$ can justifiably be treated as a (uncertainty-regularized) regression problem.
- How do local Lipschitz constants (computable perhaps for the cart-pole system) of the learned dynamics $f$ vary as uncertainty is increasingly penalized?
- Is some degree of data preprocessing (e.g., whitening) necessary to make squared (Euclidean) distance the right distance to penalize?

**Quality Of The Limitations Section:**

Additional details required

**Questions For Rebuttal:**

- How were the range of $\sigma$ values (noted in the Appendix, typically ending at 0.01) selected in practice, and what happens if you take $\sigma$ too low (e.g., do you have to compensate by lowering $\beta$)?

**Robotics Focus:**

Sufficient demonstration on hardware

**Summary Of Paper:**

In this paper the authors (i) propose an epistemic uncertainty metric for learned dynamics models, namely smoothed distance to data, (ii) identify a connection to score-based generative modeling that enables tractable gradient-based optimization with objectives incorporating this metric, and (iii) apply this methodology in the context of offline model-based reinforcement learning in a novel algorithm called Score-Guided Planning (SGP). Because the uncertainty metric is amenable to gradient-based optimization, SGP scales to problems with high dimensional state/observation spaces (e.g., images) where zeroth-order optimization methods struggle. Experiments demonstrate that SGP outperforms other baseline uncertainty estimators and optimizers (e.g., ensembles, CEM) on standard benchmarks, and moreover succeeds in high-dimensional and data-sparse regimes where it also makes more accurate policy rollout predictions than vanilla (unregularized) MBRL.

**Summary Of Recommendation:**

I recommend this paper's acceptance at CORL 2023 as the offline RL problems it addresses (particularly those with high-dimensional perceptual input) as well as the methodology it develops are highly topical in the robot learning community today. As noted above, I believe that some of its claimed insights could be better explained/better probed by additional experiments, but such additions would only be supplementary to the existing sound core work.

---

> ### Author Response · Authors · 2023-08-15
>
> Dear Reviewer,
>
> As the discussion period is almost over, please let us know if you find your concerns adequately addressed!
>
> Best, Authors

---

### Official Review · Reviewer_X384 · 2023-07-19

**Confidence:** 4
**Originality:** Very Good
**Technical Quality:** Very Good
**Clarity Of Presentation:** Very Good
**Impact:** 4

**Recommendation:**

Weak Accept: I recommend accepting the paper, but will not argue for my recommendation if the majority of other reviewers have a different opinion.

**Review:**

## Strengths

1. The core idea of using the score function as a regularizer for offline model based RL is interesting and provides a nice connection between generative modeling and offline RL.

2. The experiments demonstrate qualitatively that the method is able to work effectively with high dimensional observation and action spaces.

3. The examples about the potential weaknesses of ensemble-based uncertainty provide a nice illustration of why something like distance to data may be preferred in practical scenarios.

## Weaknesses

1. It would be good to present some more quantitative results in the experiments section. The qualitative results for the non-D4RL experiments are useful, but do not make clear quantitative comparisons between methods.

2. The performance on the D4RL benchmark suite is not particularly strong. SGP gets some small wins on some D4RL datasets, but does so poorly on others that the average performance is worse than the baselines. It could also be interesting to see comparisons on some of the higher dimensional robotic manipulation tasks in the D4RL suite instead of just the mujoco locomotion tasks.

**Quality Of The Limitations Section:**

Limitations are addressed clearly

**Questions For Rebuttal:**

See weaknesses above.

Also, the paper explains how diffuser can be seen as a relaxation of SGP. But, what (if any) are the key differences between SGP and diffuser?

**Robotics Focus:**

Sufficient demonstration on hardware

**Summary Of Paper:**

This paper presents a method for offline model-based RL (MBRL) called score-guided planning (SGP). The method is introduced by suggesting that the distance to data is good uncertainty metric and that a smoothed version of this distance is just the data likelihood. The method then uses the score function of the data generating distribution as a regularizer to prevent a gradient-based (single shooting) planner from choosing OOD actions. Experiments are presented in simple simulated control problems, a single integrator with a pixel-based action space, and a real robot pushing task.

**Summary Of Recommendation:**

Overall, this paper presented an interesting idea and backed it up with a diverse set of experiments, including on robotic tasks. There is room for improvement, but I think the paper is a good contribution and should be accepted.

---

### Official Review · Reviewer_Esgu · 2023-07-19

**Confidence:** 2
**Originality:** Very Good
**Technical Quality:** Very Good
**Clarity Of Presentation:** Very Good
**Impact:** 4

**Recommendation:**

Weak Accept: I recommend accepting the paper, but will not argue for my recommendation if the majority of other reviewers have a different opinion.

**Review:**

The paper is well-written. Technical novelty is sound, with highly relevant results to robotics. The work is interesting.

**Quality Of The Limitations Section:**

Limitations are addressed clearly

**Questions For Rebuttal:**

- In Table 1, why other methods do not have the confidence interval?
- In Pixel Single Integrator, Box Pushing with Marker Dynamics tasks, what is the scale of \beta tried after you find the best one?
- Why BO, MOPO, CQL are not discussed in the related works?


**Robotics Focus:**

Sufficient demonstration on hardware

**Summary Of Paper:**

Offline RL needs careful consideration of uncertainty estimation. This paper examines the use of smoothed distance to data as a metric, demonstrating its ability to stably converge to data and facilitate analysis of model bias through Lipschitz constants. Additionally, the authors establish an equivalence between smoothed distance to data and data likelihood, enabling the application of score-matching techniques to learn distance gradients to data. The authors highlight that offline model-based policy search problems that aim to maximize data likelihood do not require likelihood values but solely rely on the log-likelihood gradient (the score function). To tackle high-dimensional problems where zeroth-order methods struggle to scale and ensembles fail to overcome local minima, the authors propose Score-Guided Planning (SGP). This offline RL planning algorithm leverages score-matching to enable first-order planning. Empirical results show comparable performance with baselines.

**Summary Of Recommendation:**

The paper is carefully written with robotics experiments. Source code provided. I choose to accept the paper.

---

### Official Review · Reviewer_bqCh · 2023-07-23

**Confidence:** 3
**Originality:** Good
**Technical Quality:** Good
**Clarity Of Presentation:** Good
**Impact:** 2

**Recommendation:**

Strong Accept: I recommend accepting the paper and will argue for my recommendation even if other reviewers hold a different opinion.

**Review:**

## clarity

the paper is mostly written clearly.

## strengths

the work is well motivated

theoretical result connecting the smooth distance to data and the proposed score estimation

robotics experiments

## weaknesses

the way that uncertainty is treated here, it doesn;'t extrapolate.

**Quality Of The Limitations Section:**

Limitations are addressed clearly

**Questions For Rebuttal:**

line 97, take a look at [1] and cite, since they deal with exactly what you are describing (uncertainty penalty, aleatoric/epistemic separation, ensembles), PETS doesn't take into account uncertainty.

proposition 1 - this statement is almost trivial and well-known? Why is it necessary to state it here formally? And also to give a proof?

proposition 2 - in which sense is this Lischitz constant "local"? How do you define this locality? Also is it defined in the classical sense $\| x - y \| \le L \| f(x) - f(y) \|$? Details seem to be missing here.


for the pairwise finite slopes estimation of the Lipschitz constant, if I understand correctly, it doesn't seem to be "local". Can you elaborate more on the estimation of the constant?

definition 3.2 - isn't this just a mixture of Gaussians with equal weghts and variances where the means are centered at the data?

line 156 - in which sense there are no guarantees on the quality o the gradients of the likelihood function? Can you clarify this statement? If I am able to estimate the true log-likelihood, then I have the true grad log p, what does "quality of gradients" mean in this context?

eq. 9 - so you are actually modelling the mixture of Gaussians  with the smoothing parameter $\sigma$ then you start increasing it? Shouldn't the sampling distribution of x' at some point have 0 mean and variance depending on the variance schedule? As per your definition, the sampling distribution of x' is always centered on x.

line 175 - I think an algorithm box would be useful here, it is easy to glide over the fact that this method is only useful in combination with a gradient of the return. Moreover, it is not clear from the writeup how do you combine the gradients and what this corresponds to - is it just a heuristical addition with a hyperparameter? Why is this a good quality gradient that is resulting from this computation? This is a very important part of your method that is not explained well.


line 178 - "projected into space of decision variables", I don't undertand this part of the sentence. Isn't the x that you talk about in the previous section equivalent to (x,u) here? So you are modelling the perturbed joint distribution ofx,u


table 1, experiments - you method tries to stay close to the data distribution, therefore perhaps a more suitable offline algorithm for comparison would be AWAC

line 238 - subtleties*

conclusion - you state that you have not investigated aleatoric uncertainty, however, your epistemic uncertainty estimate is also questionable, since you don't take into account model extrapolation, only distance to the data. This means that your model can perfectly extrapolate, but your method would penalize it nevertheless. This I believe is the advantage of ensemble methods.

[1]  Vlastelica, Marin, Sebastian Blaes, Cristina Pinneri, and Georg Martius. "Risk-averse zero-order trajectory optimization." In 5th Annual Conference on Robot Learning. 2021.

**Robotics Focus:**

Sufficient demonstration on hardware

**Summary Of Paper:**

The paper proposes modelling the score function of the perturbed empirical data distribution in order to capture "guide" the optimization closer to the data, which is a way of tackling uncertainty in planning methods. Further as a theoreticall result i is shown that the NLL of a perturbed empirical distribution is equivalent to a smoothed distance to the data up to a factor. Results are provided in simulated domains and on real hardware.

**Summary Of Recommendation:**

there are two reasons for this rating:
1) I doubt that this method is practical, since there is no extrapolation beyond data (could be convinced otherwise)
2) unclarities (see review)

---

### Decision · Program_Chairs · 2023-08-30

**Decision:**

Accept (Poster)

**Comment:**

This paper does model-based offline RL with a score matching regularization term that encourages the resulting trajectories to go towards the offline data. All reviewers have voted for acceptance after the authors' rebuttal. While the method is interesting, one of the main questions that has remained unanswered is why are the D4RL results not always competitive with existing imitation learning or offline RL methods? Also, one of the main advantages of using offline RL is to leverage the "stitching" behavior (available data from A -> B, B->C means that through offline RL the policy can figure out A->C even if it is not in the data). The loss function here forces the trajectories to stay close to the data, so it is possible that that hinders stitching behavior, which makes me wonder why is offline RL the right RL setting to showcase this method, i.e. why not online RL + the regularization term? Finally, I would definitely have expected that the paper cite and compare against COMBO https://arxiv.org/abs/2102.08363, which as far as I can tell it doesn't.

So, while I think I would personally argue for a rejection, as I don't think offline RL is the right setting for this contribution, I will respect the reviewers evaluations and recommend acceptance.